



# On the magnitude and uncertainties of global and regional soil organic carbon: A comparative analysis using multiple estimates

Ziqi Lin[1], Yongjiu Dai[1], Umakant Mishra[2], Guocheng Wang[3], Wei Shangguan[1], Wen Zhang[3], Zhangcai Qin[1]

[1]School of Atmospheric Sciences, Guangdong Province Key Laboratory for Climate Change and Natural Disaster Studies, Sun Yat-sen University, and Southern Marine Science and Engineering Guangdong Laboratory (Zhuhai), Zhuhai 519082, China

[2]Computational Biology & Biophysics, Sandia National Laboratories, Livermore, CA 94550, United States

[3]LAPC, Institute of Atmospheric Physics, Chinese Academy of Sciences, Beijing 100029, China

*Correspondence to*: Zhangcai Qin (qinzhangcai@mail.sysu.edu.cn)

**Abstract.** Globally, soil is one of the largest terrestrial carbon reservoirs, with soil organic carbon (SOC) regulating overall soil carbon dynamics. Robust quantification of SOC stocks in existing global observation-based estimates avails accurate predictions in carbon climate feedbacks and future climate trends. In this study, we investigated global and regional SOC estimates, based on five widely used global gridded SOC datasets (HWSD, WISE30sec, GSDE, SoilGrids250m, and

GSOCmap), a regional permafrost dataset from Mishra et al. (UM2021), and a global-scale soil profile database (the World Soil Information Service soil profile database, WoSIS) reporting measurements of a series physical and chemical edaphic attributes. Our comparative analyses show that the magnitude and distribution of SOC varies widely among datasets, with certain datasets showing region-specific robustness. At the global scale, the magnitude of SOC stocks simulated by GSDE, GSOCmap, and WISE30sec are comparable, while estimates of SoilGrids250m and HWSD are at the upper and lower ends,

respectively. Global SOC stocks ranged from 577-1171 Pg C and 1086-2678 Pg C at 0-30 cm and 0-100 cm depth. The spatial distribution of SOC stocks varies greatly among datasets, especially in the northern circumpolar and Tibetan Plateau permafrost regions. In general, the UM2021 and WISE30sec perform better in the northern circumpolar permafrost regions, and GSDE performs better in China. SOC stocks estimated by different datasets also show large variabilities across different soil layers and biomes. Overall, GSOCmap performs well at 0-30 cm depth, while SoilGrids250m and GSDE perform better at multiple

depths. Among the five gridded global datasets, SoilGrids250m exhibits a more consistent spatial pattern and depth distribution with WoSIS. Large uncertainties in existing global gridded SOC estimates are generally derived from soil sampling density, diverse sources and mapping methods for soil datasets. We call for future efforts for standardizing soil sampling efforts, cross-dataset comparison, proper validation, and overall global collaboration to improve SOC estimates. The data are available at https://doi.org/10.6084/m9.figshare.20220234 (Lin et al., 2022).



## 1 Introduction


Soil stores twice the amount of carbon in atmosphere and vegetation combined thus plays a fundamental role in the global

carbon cycle (Piao et al., 2009; Bastida et al., 2019). Soil carbon consists of inorganic carbon (SIC) and organic carbon (SOC),

and the latter accounts for over 60% of total soil carbon pool (Lal, 2004; Batjes, 1996; Houghton, 2007). Carbon enters soil

profiles through organic inputs and leaves soil *via* mainly heterotrophic respiration, both processes can be significantly affected

by natural and anthropogenic perturbations (Yang et al., 2020; Lorenz and Lal, 2022). Due to its large size, a minor change in

SOC stock can have profound impacts on atmospheric $CO_2$ concentration and hence climate change (Ciais et al., 2013; Köchy

et al., 2015b). Thus, accurate estimation of SOC is essential for carbon climate feedbacks and future climate change projections.

In recent decades, there have been many soil data products, at either global or regional scale, constructed to satisfy diversified

demands for soil information (Batjes et al., 2017). At regional and national scales, there are many datasets compiled by local

governments and regional organizations based on regional soil surveys, e.g., the Soil and Terrain databases (SOTER) (van

Engelen and Dijkshoorn, 2013), the European Soil Database (ESDB) (Panagos et al., 2012), the 1:100 million scale Soil Map

of China (Shi et al., 2004), the Australian Soil Resource Information System polygon attributed surface (ASRIS) (McKenzie

et al., 2000), and Africa Soil Information Service (AfSIS) (Leenaars et al., 2014). The global soil datasets are primarily

generated by international institutions and organizations, like the Food and Agriculture Organization of the United Nations

(FAO). These organizations first collected soil information from the national and/or regional soil databases, then processed

and compiled the information to construct a harmonized global soil dataset. For example, FAO integrated nearly 600 soil maps

around the world to compile the World Soil Map (SMW) and then digitized it to build the Digital World Soil Map (DSMW)

(FAO, 1974, 1988, 1995). The Harmonized World Soil Database (HWSD) was established by using the regional and national

soil information assimilated and harmonized by International Institute for Applied Systems Analysis (IIASA) and FAO (FAO

et al., 2012).

So far, a number of global and regional soil datasets have been produced and used, but they often vary largely in terms

of data sources, mapping methods, soil properties, and the purposes and scope. For instance, the ISRIC-WISE-derived soil

property databases (Batjes, 2005, 2012, 2016) and the SoilGrids system (Hengl et al., 2014, 2017) differ in data sources and

mapping approaches, but they have all been widely used for different studies. The Global Soil Dataset for Earth System Models

(GSDE) (Shangguan et al. 2014) was originally developed for Earth System Models (ESM), and the many soil properties

included can well serve as model inputs (Shangguan et al. 2014). By contrast, the Global Soil Organic Carbon Map (GSOCmap)

was established to simulate the distribution of global soil organic carbon only (FAO and ITPS, 2018). It should also be noted

that many soil data products have been updated from previous soil databases, to deal with problems including data outdating,

coarse spatial resolution, and differences in process of soil data. For example, HWSD started on the basis of the framework of

DSMW (FAO et al., 2012), complemented soil information, and eventually superseded DSMW. ISRIC-World Soil Information

further used the soil profiles from Harmonized continental SOTER-derived database (SOTWIS) (van Engelen et al., 2005) and

the International Soil Carbon Network (ISCN) (Nave et al., 2015) to supplement the data missing area of HWSD and

constructed WISE30sec.

Previous studies have evaluated SOC estimates of various soil data products from different perspectives. Scharlemann et

al. (2014) performed a meta-analysis and reported that global 0-1 m SOC stocks ranged from 504 to 3000 Pg C, with a median

of 1460.5 Pg C. Dai et al. (2019) evaluated several existing soil datasets in the application of ESMs. To construct or improve

a certain soil data product, many inter-comparisons among different SOC estimates have been done as references (Hiederer

and Köchy, 2011; Köchy et al., 2015a; FAO and ITPS, 2018). However, few studies have comprehensively compared and

assessed SOC estimates across multiple existing products, identifying knowledge gaps in global and regional SOC magnitude

and uncertainties that may differ among datasets. It has been recognized that the high variability in SOC estimates among

different studies could be from diverse calculation approaches (Köchy et al., 2015a). Moreover, current global soil datasets are

updated frequently and there is no unified standard to assess the accuracy of global soil datasets with different formats, content

and resolution. If the uncertainty in SOC estimates is to be reduced, the differences among the databases and the impact of

these differences on the SOC estimates should be quantified.

In this study, we used five global soil datasets (HWSD, WISE30sec, GSDE, SoilGrids250m and GSOCmap) to quantify

the magnitude and distribution of global SOC stocks and its uncertainties at depth of 0-30 cm and 0-100 cm, respectively. We

used a regional permafrost dataset developed by Mishra et al. (2021) (called UM2021 in this study) to facilitate comparison in

the regions with permafrost. We also used the World Soil Information Service soil profile database (WoSIS) as a reference for

comparison purpose. Through the robust quantification of uncertainty in existing global SOC observation-based estimates, we

can quantify the magnitude and distribution of SOC stocks among current estimates, then identify areas for future

improvements, which could reduce the uncertainty in model projected SOC dynamics, carbon climate feedbacks, and future

climate trends.

## 2 Methods and materials

### 2.1 Soil datasets

In this study, we used five widely used and recently updated global soil datasets with estimates of SOC content or density, i.e.,

HWSDv1.21 (FAO et al., 2012), WISE30sec (Batjes, 2015, 2016), GSDE (Shangguan et al., 2014), SoilGrids250mv1.0 (Hengl

et al., 2017) and GSOCmapv1.5 (FAO and ITPS, 2018) (Table 1). Specifically, the HWSD is one of the most coherent and

widely used global soil databases, developed by FAO, IIASA, ISRIC-World Soil Information (ISRIC), Institute of Soil Science,

Chinese Academy of Sciences (ISSCAS), and Joint Research Centre of the European Commission (JRC). It was produced by

linking the regional and national soil properties information to the soil map of the world according to taxonomy-based

pedotransfer functions. The GSDE and WISE30sec are the improvements over the HWSD. They expanded the spatial coverage

of soil maps and added more soil profiles. Specifically, using a framework of HWSD, the GSDE supplemented soil information

from the U.S. General Soil Map (GSM) (USDA-NCSS, 2006), the Soil Landscapes of Canada (SLC, version 3.2) (Soil

Landscapes of Canada Working Group, 2010), ASRIS, the soil database of China for land surface modeling (Shangguan et al.,

2013), and the SOTWIS of the Indo-Gangetic Plains (Batjes et al., 2004), Jordan (Batjes et al., 2003), and Kenya (Batjes and

Gicheru, 2004). To satisfy the needs for ESMs inputs, the GSDE included 34 soil properties for 8 depth intervals (up to 2.3 m

depth). WISE30sec corrected the HWSD by using the Köppen-Geiger climate zone map as the categorical covariate and

complemented with about 8000 soil profiles from some northern high latitude regions, and estimated 20 soil properties of 7

layers to a depth of 2 m. For SoilGrids250m, the first version (1.0) was still used here for method consistency between 0-30

cm and 0-100 cm. The latest version (SoilGrids 2.0) was recently released (Poggio et al., 2021), but the data for 0-100 cm SOC

stocks is unavailable at the moment, and re-analysis by users to derive 0-100 cm SOC stocks could potentially result in

inconsistent results (personal communication). SoilGrids250m (v1.0) was mapped by machine learning methods based on

about 150000 soil profiles from the WoSIS database and 158 environmental covariates (Hengl et al., 2017). It estimated 11

soil properties for 6 depth intervals (up to 2 m depth) at 250 m spatial resolution. The GSOCmap is a country-specific grid soil

database for 0-30 cm SOC density, developed by the Global Soil Partnership (GSP). The GSP collected national SOC data

generated by the member countries according to the harmonized standards and then filled the gaps by using publicly available

data or simulations.

Moreover, we used a regional permafrost dataset from Mishra et al. (2021) (called UM2021 in this study) for further

comparison of the SOC stock in the permafrost affected soils, both in northern circumpolar region and the Tibetan Plateau.

The UM2021 was created by combining over 2700 soil profiles with environmental variables in a geospatial framework. It

provided the most up-to-date SOC density information and associated uncertainty estimates of permafrost affected soils for 4

depth intervals at a spatial resolution of 250 m.



**Table 1 Key features of soil datasets**

| Name (version) | HWSD (Version 1.21) | GSDE | WISE30sec | SoilGrids250m (Version 1.0) | GSOCmap (Version 1.5) | UM2021 |
|---|---|---|---|---|---|---|
| Number of layers | 2 | 8 | 7 | 6 | 1 | 4 |
| Depth interval (cm) | 30, 100 | 4.5, 9.1, 16.6, 28.9, 49.3, 82.9, 138.3, 229.6 | 20, 40, 60, 80, 100, 150, 200 | 5, 15, 30, 60, 100, 200 | 30 | 30, 100, 200, 300 |
| Properties | 16 | 34 | 20 | 11 | 1 | 1 |
| Spatial Resolution | 30″ (~1km) | 30″ (~1km) | 30″ (~1km) | 250m | 30″ (~1km) | 250m |
| Update time | 2012 | 2014 | 2016 | 2017 | 2018 | 2021 |
| Domain | Global | Global | Global | Global | Global | The northern circumpolar region and the Tibetan Plateau |
| Data access | http://webarchive.iiasa.ac.at/Research/LUC/External-World-soil-database/HTML/HWSD_Data.html?sb=4 (last access: 5 July 2022) | https://data.isric.org/geonetwork/srv/eng/catalog.search#/metadata/dc7b283a-8f19-45e1-aaed-e9bd515119bc (last access: 5 July 2022) | http://globalchange.bnu.edu.cn/research/soilw (last access: 5 July 2022) | https://files.isric.org/soilgrids/former/2017-03-10/data/ (last access: 5 July 2022) | http://54.229.242.119/GSOCmap/ (last access: 5 July 2022) | https://datadryad.org/stash/dataset/doi:10.7941/D1GD1H (last access: 5 July 2022) |
| Reference | FAO et al. (2012) | Shangguan et al. (2014) | Batjes. (2016) | Hengl et al. (2017) | FAO and ITPS. (2018) | Mishra et al. (2021) |
| DOI | NA | https://doi.org/10.1002/2013MS000293 | https://doi.org/10.1371/journal.pone.0169748 | https://doi.org/10.1016/j.geoderma.2016.01.034 | NA | https://doi.org/10.1126/sciadv.aaz5236 |

Notes: NA denotes not available.

**2.2 Analysis**

To make these datasets comparable, we transformed all datasets to the same coordinate system (WGS1984), calculated SOC

density for each layer using the following Eq. (1) and resampled them to a spatial resolution of 30″×30″. The transforming,

calculating and resampling can be done in ArcGIS10.4 or using open access alternatives (e.g., QGIS, R).

$$SOC_D = SOC_C \times BD \times D \times (1 - CF), \tag{1}$$

where $SOC_D$ is SOC density (t ha$^{-1}$), $SOC_C$ is SOC content (% weight), $BD$ is soil bulk density (g cm$^{-3}$), $D$ is the depth of soil

layers (cm), and $CF$ is the coarse fragments (% weight). Due to the different layer schemes of these soil datasets, we figured

up SOC density for the depth of 0-30 cm and 0-100 cm by Eq. (2) and (3):

$$SOC_{Di} = \begin{cases} SOC_{Di}, & b \leq m \\ SOC_{Di} \times \frac{m-a}{b-a}, & b > m \end{cases}, (m = 30 \ or \ 100), \tag{2}$$





$$SOC_{D(0-m)} = \sum_{i=1}^{n} SOC_{Di}, \tag{3}$$

where $SOC_{Di}$ is SOC density (t ha$^{-1}$) for each layer, $SOC_{D(0-m)}$ is SOC density (t ha$^{-1}$) at 0-30 cm or 0-100 cm depth, $m$ is the

target depth (30 cm or 100 cm), $a$ is the upper depth (cm) of layer, $b$ is the bottom depth (cm) of layer, $n$ is the number of

soil layers. In each grid, the uncertainty of SOC$_D$ induced by data source (derived from above-mentioned five datasets) is

expressed as coefficient of variation for each grid cell by Eq. (4):

$$CV = \frac{\sqrt{\frac{1}{N}\sum_{i=1}^{N}(X_i-\bar{X})^2}}{\bar{X}}, \tag{4}$$

where $CV$ is the coefficient of variation, $X_i$ is SOC density in each dataset, $\bar{X}$ is the average of all soil datasets, $N$ is the total

number of the soil datasets.

In the study, soil profiles from the WoSIS were used as a reference to evaluate the uncertainties of the five global soil

datasets, and the global land cover data from the MODIS Land Cover Climate Modeling Grid Product (MCD12C1) (Friedl

and Sulla-Menashe, 2015) was used for the evaluation of uncertainties of SOC estimates among biomes. The WoSIS global

soil profile database was developed by ISRIC and comprised of a large amount of quality-assessed and standardized soil

profiles (about 196498 profiles in the latest version) from 173 countries (Batjes et al., 2020). The MCD21C1 supplied global

maps of 17 land cover classes based on the International Geosphere-Biosphere Programme's classification schemes (Friedl and

Sulla-Menashe, 2015), and we categorized the global land surfaces into 8 major biomes: forests, shrublands/savannas,

grasslands, croplands, barren lands, urban and built-up lands and permanent wetlands. The lands without soil cover, like water

bodies and permanent snow and ice, were not discussed in this study. Here, we represented the comparative results between

the WoSIS and other soil datasets using the Taylor diagram, a concise graph that can summarize how closely a model matches

the observations. In this diagram, the spatial correlation coefficients and the root mean square errors (RMSE) between

simulated and observed fields, as well as the normalized standard deviations of the simulated value from the global mean, can

all be shown by a point in the polar coordinate system (Taylor, 2001, 2005; Hu et al., 2022). The Taylor Diagram can be

plotted using MATLAB file from Guillaume (2022) or other open access scripts like R.

## 3 Results

### 3.1 Global SOC distribution and stocks

On the global scale, the spatial distributions of 0-30 cm SOC density from the five global datasets are generally consistent,

with values increasing from lower to higher latitudes (Fig. 1 and S1a). Higher SOC density is concentrated in the northern

high latitudes including Russia, Northern Europe, Alaska, and northern Canada, along with some equatorial regions between

10° N to 10° S and southern South America. Relatively lower SOC density is found in the mid and low latitudes ranging from

10° N-50° N and 10° S-40° S, such as northern and southern Africa, central and western Asia, and Australia. Note that in these
datasets, there are soil information gaps in some areas, including Greenland in the GSOCmap (Fig. 1e), and some regions in

the Sahara Desert in the HWSD (Fig. 1a), WISE30sec (Fig. 1b), and GSDE (Fig. 1c). At the depth of 0-100 cm, the distributions

of SOC density from the five soil datasets share a consistent pattern with those of 0-30 cm SOC density, with higher values in

the northern high latitudes and certain equatorial regions (Fig. S1b and S2). For global SOC stocks, the estimates from the five

soil datasets are in a range of 577-1171 Pg and 1086-2678 Pg with averages of 828 and 1873 Pg, for 0-30 cm 0-100 cm soil

depths, respectively (Fig. 1f). Among the five datasets, the SOC stocks estimated by the Soilgrids250m is the highest and that

estimated by the HWSD is the lowest. The estimates provided by the WISE30sec and GSDE are relatively comparable.

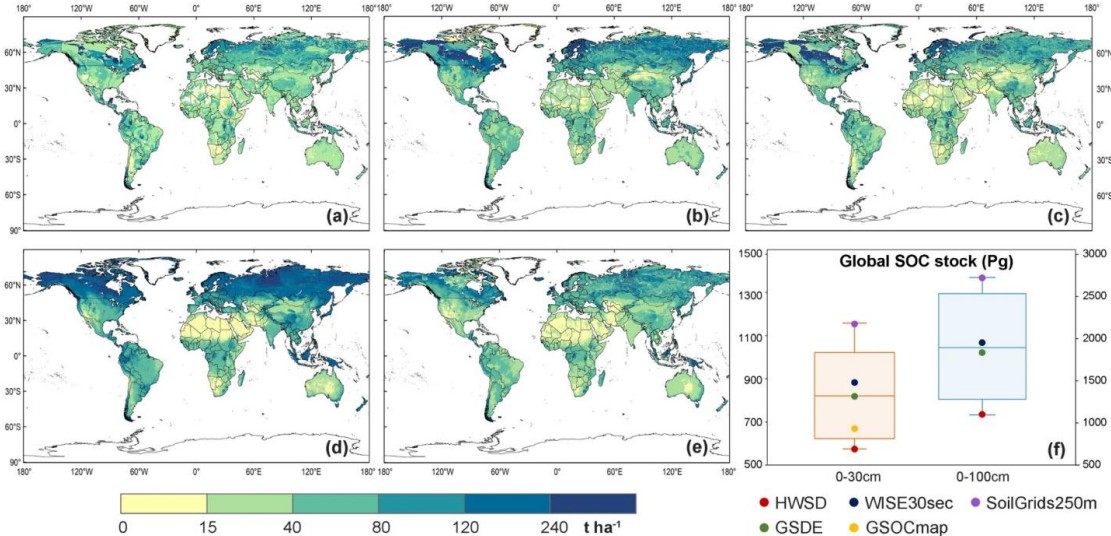

**Figure 1: Global SOC density and stocks vary by datasets. (a-e) SOC density of 0-30 cm (t ha$^{-1}$) based on HWSD, WISE30sec, GSDE, SoilGrids250m, and GSOCmap, respectively. (f) Global SOC stocks (Pg) estimated for five datasets.**

**3.2 Global SOC density differences among datasets**

Although the five datasets show similar patterns of SOC density distribution, their magnitudes of SOC density differ globally

and in specific regions. The coefficient of variations (CV) range from 0.15% to 179% for 0-30 cm SOC density (Fig. 2a) and

from 0.11% to 163% for 0-100 cm SOC density (Fig. 2b), illustrating the spatial heterogeneity of SOC density among the five

datasets. The major differences among the five datasets are observed in the northern circumpolar region and the Tibetan Plateau,

where the most permafrost soils are located. There are also differences in Southeast Asia, some areas of the Sahara Desert, the

basin of the upper White Nile, the Great Basin of Australia and some valleys around the Cordillera. Meanwhile, in these regions,

the five datasets differ more at 0-100 cm depth than at 0-30 cm depth.



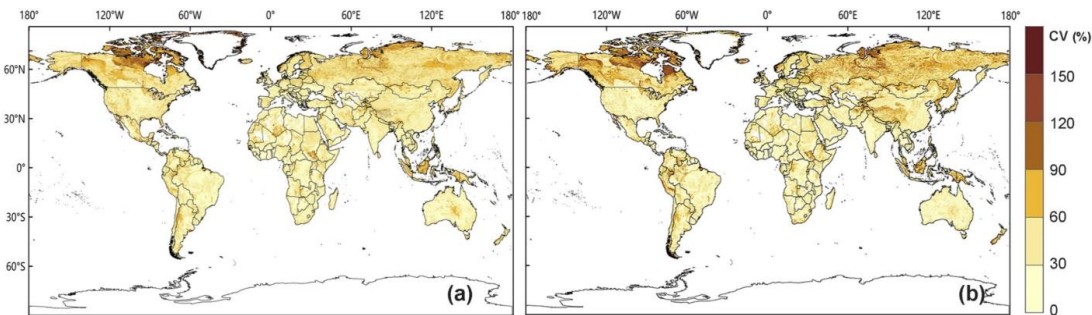

**Figure 2: The coefficient of variations (CV, %) estimated for (a) 0-30 cm and (b) 0-100 cm SOC density (t ha$^{-1}$) using five datasets.**

The differences among datasets can be further evidenced by using the mean of the five datasets as the benchmark.

Generally, for 0-30 cm SOC density, the HWSD has negative biases (Fig. 3a) and the SoilGrids250m has positive biases (Fig. 3d). The WISE30sec, GSDE and GSOCmap are relatively close to the mean in the spatial distribution of SOC density (Fig. 3b, 3c and 3e). Specifically, in the northern circumpolar region, where the differences are significant, the WISE30sec and SoilGrids250m have positive deviations from the mean, while the HWSD, GSDE and GSOCmap have negative deviations. The magnitude of SOC density estimated by the SoilGrids250m is the highest in most areas of world, followed by the

WISE30sec, GSDE and GSOCmap, and that estimated by the HWSD is the lowest (Fig. S3). However, in Northern Europe and Northwest Canada, the SoilGrids250m is 100 t ha$^{-1}$ lower at least than the WISE30sec and GSDE, while it is higher than the HWSD and GSOCmap (Fig. S3). In Southeast Asia, the biases of the SoilGrids250m and the GSOCmap from the mean are positive, while those of the HWSD, WISE30sec and GSDE are negative. The SOC density estimate of the SoilGrids250m is 50-300 t ha$^{-1}$ larger than that of the GSOCmap, and the estimates of the WISE30sec and GSDE are similar, both higher than

that of the HWSD (Fig. S3).

Furthermore, the relative differences between the individual dataset and the mean magnify the details of discrepancy among the datasets on the regional scale (Fig. 3f-j). In the parts of northern Africa and the Middle East, the HWSD is above the mean and the SoilGrids250m is below the mean, opposite to the biases in other regions. In central Australia, the HWSD and WISE30sec have positive deviations from the mean and others have negative deviations, but the differences across the

five datasets are relatively small (within approximately ±50 t ha$^{-1}$) (Fig. S3). In the basin of the upper White Nile, the SOC density estimated by the GSDE is over 300 t ha$^{-1}$ higher than that estimated by others (Fig. S3). For 0-100 cm SOC density estimates, the pattern of differences among the soil datasets is similar with that at 0-30 cm depth, but the magnitude of biases is larger (Fig. S4).



**Figure 3: Differences of 0-30cm SOC density (t ha$^{-1}$) between individual dataset and the mean. The mean is the average of the five datasets. Left column is the absolute difference of (a) HWSD, (b) WISE30sec, (c) GSDE, (d) SoilGrids250m, and (e) GSOCmap. Right column is the relative difference of (f) HWSD, (g) WISE30sec, (h) GSDE, (i) SoilGrids250m, and (j) GSOCmap.**





In comparison with WoSIS reference, the five soil datasets generally provide a better estimate of SOC density for 0-30cm

depth than for 0-100 cm depth. The correlation coefficients of five datasets with the WoSIS range from 0.19 to 0.47 for 0-30

cm (Fig. 4a) and from 0.14 to 0.48 for 0-100 cm (Fig. 4b). The root mean square errors (RMSE) for 0-30 cm SOC density is

at the range of 0.94 to 1.28 (Fig. 4a), which is slightly smaller than that for 0-100 cm (0.98-1.48) (Fig. 4b). The normalized

standard deviation of 0-30 cm SOC density is from 0.56 to 1.03 (Fig. 4a) and that of 0-100 cm is from 0.50 to 1.23 (Fig. 4b).

The SOC density simulations from the SoilGrids250m are the closest to the observations from the WoSIS at both 0-30 cm and

0-100 cm depths (the correlation coefficients are 0.47 and 0.48, respectively). At 0-30 cm depth, the simulations from the

GSOCmap are also relatively close to the WoSIS (the correlation coefficient is 0.31). The HWSD, WISE30sec and GSDE are

weakly correlated with the WoSIS, and it is related to the different metadata sources of these soil datasets (Fig. 9). The

amplitudes of the WISE30sec and GSDE are similar to the WoSIS at 0-30 cm depth. The lower standard deviation of the

HWSD indicates that there is an underestimation of SOC density in the local region of the HWSD compared to the WoSIS.

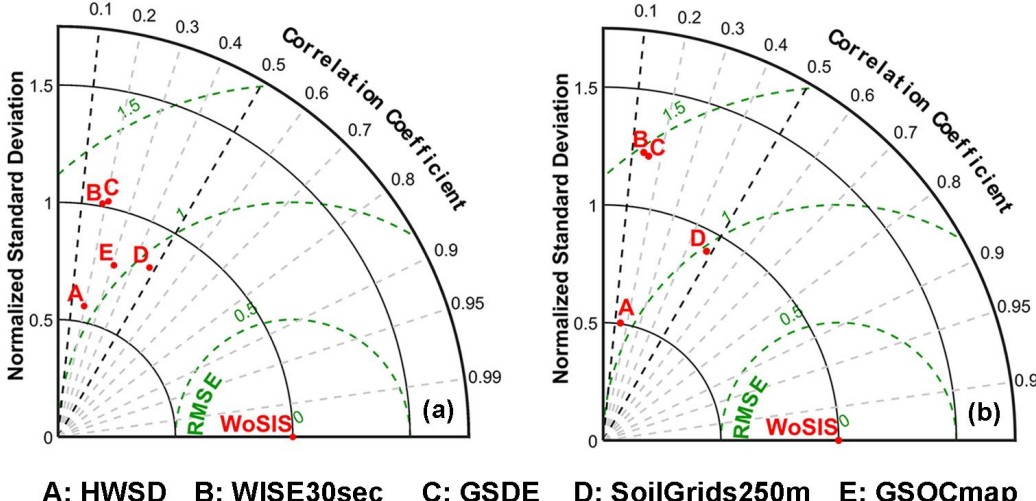

**Figure 4: Normalized Taylor diagram of the comparison of SOC density (t ha$^{-1}$) between WoSIS and individual soil dataset at the**
**depth of (a) 0-30 cm (n=14771) and (b) 0-100 cm (n=15245). The GSOCmap only includes 0-30 cm SOC density estimates.**

**3.3 Regional SOC density differences among datasets**

The largest discrepancy among the global soil datasets is found in the northern circumpolar region and the Tibetan Plateau,

where most permafrost-affected soils are located (Fig. 2). The UM2021, the regional permafrost dataset reported by Mishra et

al., (2021), is used to compare the simulations and uncertainties of the SOC density in these areas. In the northern circumpolar

permafrost region, the high simulations of 0-30 cm SOC density are concentrated in West Siberian Plain, Alaska, and from

Northwest Territories to the southern shore of Hudson Bay in Canada (Fig. 5), where the Histosol and Gleysol are widely

located. Overall, the SoilGrids250m presented the highest SOC density, followed by WISE30sec, UM2021 and GSDE. The

GSOCmap lacks soil information for Greenland Island, and its estimated SOC density is relatively higher than HWSD and

lower than GSDE. Compared with the mean of the six datasets, the SOC density estimated by UM2021 is the closest, followed

by the WISE30sec, GSDE and GSOCmap, and the SoilGrids250m and HWSD have the largest positive and negative deviation,

respectively (Fig. S5). The CVs show the differences across the datasets in the northern circumpolar permafrost region are

distributed in northern Canada (60°-80° N, 60°-120° W), the basins of Ob River and Yenisei River on the Western Siberian

Plain (60°-75° N, 60°-90° E) (Fig. S6a). Specifically, in northern Canada, SOC density estimated by the SoilGrids250m and

UM2021 are above the mean and the deviation of SoilGrids250m is larger than that of UM2021 (Fig. S5). In the Western

Siberian Plain, the discrepancy across the five datasets is mainly due to the high SOC density of SoilGrids250m which is

overall higher than the mean (>200 t ha$^{-1}$) (Fig. S5). Differences among the soil datasets for 0-100 cm SOC estimates are more

extensive for 0-30 cm SOC estimates, although their distribution patterns are similar (Fig. S6b, S7 and S8).

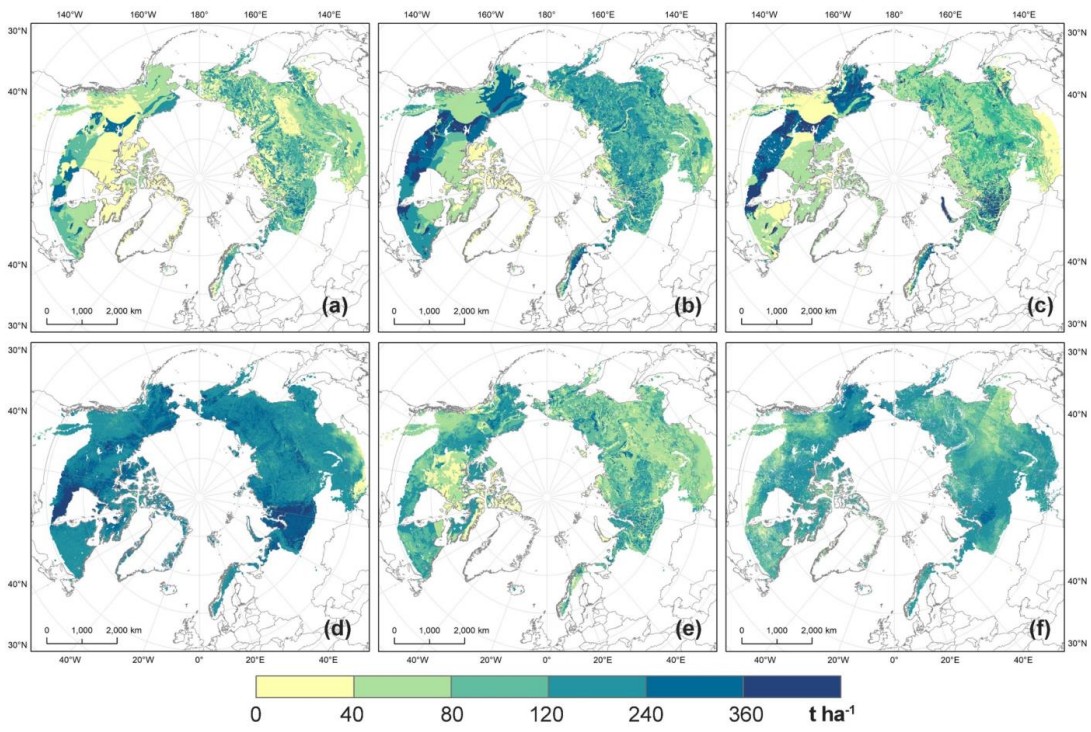

**Figure 5: Spatial distribution of 0-30 cm SOC density (t ha$^{-1}$) in the northern circumpolar permafrost region from (a)HWSD, (b)WISE30sec, (c)GSDE, (d)SoilGrids250m, (e)GSOCmap and (f) UM2021.**

In the Tibetan Plateau, the estimated 0-30 cm SOC of most datasets decreases gradually from southeast to northwest (Fig.

6a and c-f), except WISE30sec (Fig. 6b). Relatively, the SOC density simulated by GSDE, SoilGrids250m and UM2021 are

closer to the pattern of SOC distribution based on the Tibetan Plateau observations, with high values in the forest of

southeastern Tibetan Plateau and low values in the desert of northwestern Tibetan Plateau (Wang et al., 2019; Tian et al., 2008).



The large CVs among the datasets are found in the western and southeastern Tibetan Plateau, as well as the Tsaidam Basin (Fig. S9a). Compared with the mean of the six datasets, the SoilGrids250m and UM2021 has the largest positive and negative overall deviation, respectively. However, the SoilGrids250m is lower than the mean in the western Tibetan Plateau and the Tsaidam Basin, and the UM2021 is higher than the mean in the southeastern Tibetan Plateau (Fig. S10). The SOC density

simulated by HWSD and GSOCmap are relatively close to the mean in the western Tibetan Plateau and higher than the mean in the Tsaidam Basin (Fig. S10). Different from the SOC distribution patterns of other datasets, the SOC density estimated by the WISE30sec is higher than the mean in the western Tibetan Plateau, which results in the large CVs of this region (Fig. S10). For 0-100 cm SOC estimates, the differences across soil datasets are more significant for 0-30 cm SOC estimates, although their distribution patterns are comparable (Fig. S9b, S11 and S12).

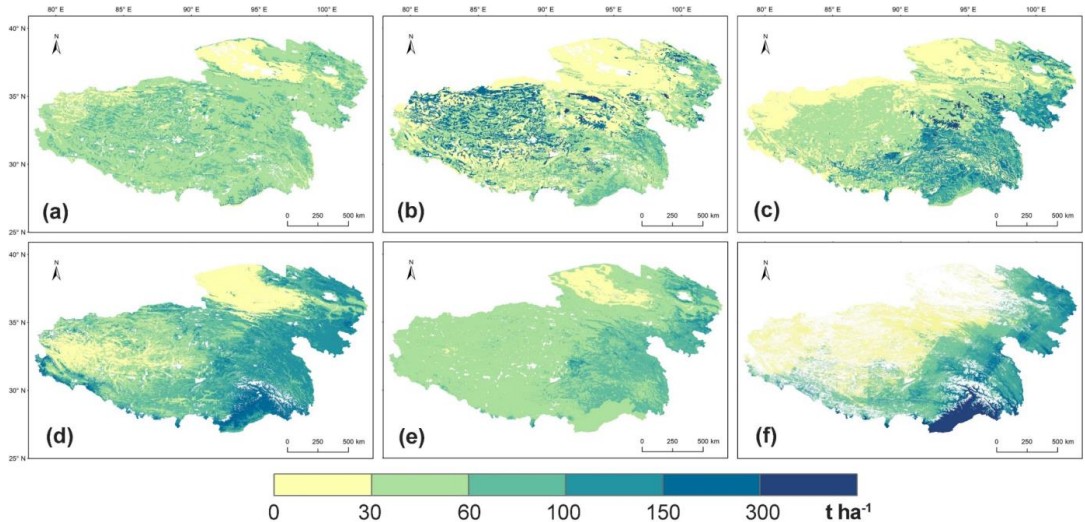


**Figure 6: Spatial distribution of 0-30 cm SOC density (t ha$^{-1}$) in the Tibetan permafrost region from (a)HWSD, (b)WISE30sec, (c)GSDE, (d)SoilGrids250m and (e)UM2021, (e)GSOCmap and (f) UM2021.**

**3.4 SOC estimates by biomes**

The Taylor diagrams indicate that the five soil datasets perform relatively better in estimating SOC density of grasslands,

croplands and shrublands/savannas than other biomes, with correlation coefficients ranging from 0.1-0.6 and normalized standard deviations of approximately 1.0 (Fig. 7). The performance is poor in the simulation of SOC density in permanent wetlands, with correlation coefficients ranging from −0.2 to 0.5 and normalized standard deviations over 2.0 (Fig. 7). For SOC estimates over forests, normalized standard deviations are below 1.0 for all five datasets, indicating that the SOC estimates from five datasets are lower than the WoSIS observations in this biome (Fig. 7). The poor performance of permanent wetlands

and forests may relate to the low sampling density of these biomes (Fig. S13). For SOC estimates in different biomes, the five global soil datasets have different performances. The SoilGrids250m shows the closest correlation to the WoSIS observations





for SOC estimates in most biomes, followed by GSOCmap, and HWSD underperforms all other datasets in most biomes (Fig.

7). In croplands and urban and built-up lands, SoilGrids250m has larger normalized standard deviations with WoSIS (0.5 of

croplands and 1.5 of urban and built-up lands, respectively), though it is still closer to WoSIS than others (Fig. 7). Compared

with SOC estimates of other biomes, the WISE30sec performs better in shrublands/savannas and urban and built-up lands, and

GSDE performs better in croplands (Fig. 7).



**Figure 7: Normalized Taylor diagrams of the comparison of 0-30 cm SOC density (t ha$^{-1}$) between WoSIS and individual soil dataset for different biomes. N is the number of WoSIS soil profiles in each biome.**

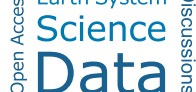

## 4 Discussion

### 4.1 Overall assessment of SOC estimates

In this study, the results indicate no consensus on global SOC estimates in these observation-based estimates, and each dataset performs differently in specific regions. Generally, in terms of both simulation quality and sampling density, the GSOCmap provides preferable global SOC estimates at 0-30 cm depth, while the SoilGrids250m and GSDE simulate the global SOC magnitudes and distribution better at both 0-30 cm and 0-100 cm depth (Fig. 8). In the northern circumpolar permafrost region, the UM2021 has the smallest uncertainty on SOC simulations, followed by the WISE30sec. In China, the GSDE offers the best SOC estimation using a large amount of soil observational data, particularly in the Tibetan Plateau. As shown in Fig. 8, for specific regions, the size of the circles indicates the discrepancies among these estimates. The larger the circle, the greater the differences in the data quality or sampling density in this area for the datasets involved in the comparison. For instance, in the northern circumpolar permafrost region (region 4 in Fig. 8), the soil sampling density is much lower than in other regions due to the variation in the extent of the permafrost region and the harsh climatic conditions, which affects the prediction of the SOC spatial heterogeneity and the uncertainty of measurements (Köchy et al., 2015a; Mishra et al., 2013, 2021). In China (region 1 in Fig. 8) and Africa (region 6 in Fig. 8), the differences among the five datasets are relatively smaller in data quality and sampling density dimensions, while the variability in Southeast Asia (region 2 in Fig. 8) is larger. Compared with other datasets, the SoilGrids250m has greater SOC estimates in most high latitudes and some tropical islands, like Sri Lanka (Vitharana et al., 2019), though it is very close to the observations from the WoSIS. However, these datasets have advantages and limitations in different aspects of simulation. For example, the GSDE has most abundant soil information, with 34 soil properties up to 2.3 m depth, which can satisfy the diversity of applications (e.g., modeling). The GSOCmap specifically targets at 0-30 cm SOC data, without capability to reflect more soil properties and soil carbon information at variable depth intervals (Table S1).

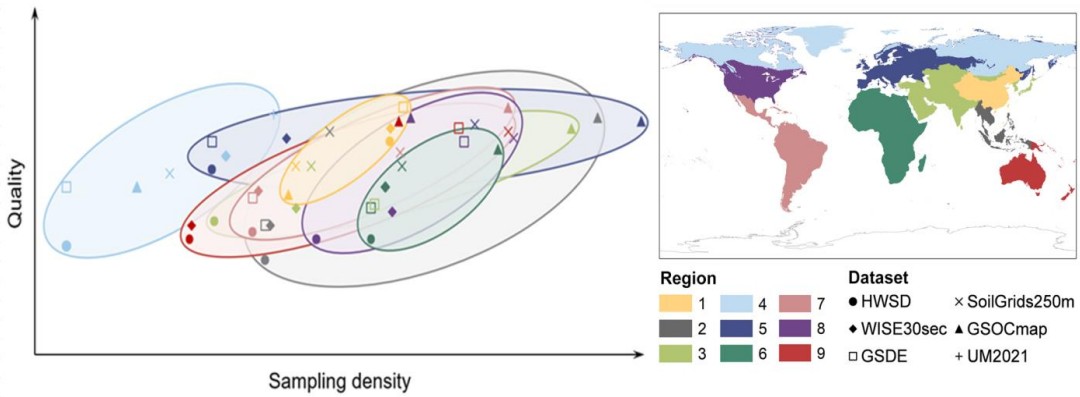

**Figure 8: A qualitative assessment of soil datasets at the regional scale. The circles with different colors represent different regions.**



### 4.2 Sources of differences and future needs

The diversity of data sources and different mapping methods are important reasons for the differences among soil datasets. The

conventional mapping approach is the knowledge-based linkage method (also known as the taxotransfer rule-based method), which links soil profiles and soil mapping units on soil type maps (Batjes, 2003; Dai et al., 2019). The HWSD, GSDE, and WISE30sec are constructed using this approach, and the filed observations these estimates used are more or less related (Fig. 9a). In recent decades, the digital soil mapping methods have been used to generate SOC estimates, and it is modeled using machine learning or other spatial interpolation approaches based on soil profiles and environmental covariates such as climate,

topography, and land use (McBratney et al., 2003; Hengl et al., 2017). The SoilGrids250m and GSOCmap both use digital soil mapping methods, and their data sources are also somewhat similar (Fig. 9b). The SOC estimates generated by digital mapping methods usually better reflect continuous spatial variation in soil properties than those using the linkage methods. Adjacent soil types in reality often have no obvious spatial boundaries, but rather there is a certain range of transition zones, and the soil properties in the transition zones are similar to the adjacent soils. However, the linkage method assigns only one statistical

value to the soil type in the soil unit, resulting in abrupt changes in attribute values at the boundaries of the soil polygon (Dai et al., 2019; Zhu et al., 2018). Moreover, the quality of field observations impacts the robustness of generated SOC estimates. For instance, the parts of the HWSD that still utilizes the DSMW such as North America, Australia, West Africa (excluding Senegal and Gambia) and South Asia are considered less reliable, while most of the areas covered by SOTWIS databases are considered to have higher reliability (FAO et al., 2012) (Fig. S14).



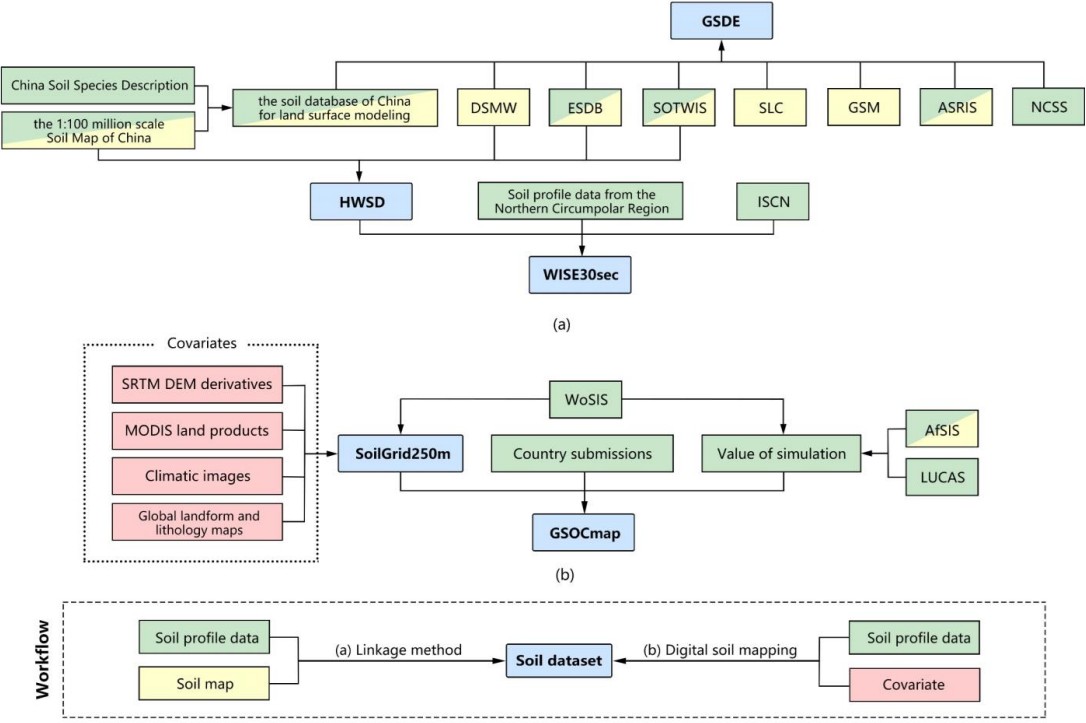


**Figure 9: Data sources of global soil datasets. The sources include DSMW (the Digital Soil Map of the World), ESDB (the European Soil Database), SOTWIS (SOTER and WISE-derived databases), ISCN (International Soil Carbon Network), SLC (the Soil Landscapes of Canada), GSM (the U.S. General Soil Map), ASRIS (the Australian Soil Resource Information System polygon attributed surface), NCSS (National Cooperative Soil Characterization Database), WoSIS (the World Soil Information Service soil**

**profile database), AfSIS (Africa Soil Information Service), LUCAS (Land Use and Coverage Area frame Survey).**

In addition to the objective disparities among SOC estimates, the different estimation approaches also contribute to diverse estimates of SOC stocks (Table. 2). For example, Köchy et al. (2015a) and Tifafi et al. (2018) calculated widely varying global SOC stocks at 0-100 cm depth, although their estimates were based on the same dataset. Since SOC estimates are dependent on numerous factors such as SOC content, bulk density, and coarse fragments, uncertainty and/or errors in measurement of one

factor may affect the final SOC stock estimation (Köchy et al., 2015a; Poeplau et al., 2017). In the estimate from Hiederer and Köchy (2011), notable differences in global SOC stock estimates were ascribed to the varying bulk density parameters.

**Table 2 Different estimates for SOC stocks at the depth of 0-100 cm and 0-30 cm**

| Depth (cm) | SOC stock (Pg C) | Data source | Reference |
|---|---|---|---|
| 0-100 | 2500 | HWSD v1.2 | (Tifafi et al., 2018) |
| | 3400 | SoilGrids250m v1.0 | (Tifafi et al., 2018) |
| | 1325 | HWSD v1.2 | (Köchy et al., 2015a) |
| | 1408±154 | WISE30sec | (Batjes, 2015) |
| | 504-3000 | 27 datasets | (Scharlemann et al., 2014) |
| | 1417 | HWSD v1.1 | (Hiederer and Köchy, 2011) |
| | 1399 | NRCS | (Hiederer and Köchy, 2011) |
| | 1459 | FAO2007 | (Hiederer and Köchy, 2011) |
| | 991 | WISE5by5min | (Hiederer and Köchy, 2011) |
| | 1206 | DSMW | (Hiederer and Köchy, 2011) |
| | 1500 | 16 estimates | (Amundson, 2001) |
| | 1462-1548 | WISE | (Batjes, 1996) |
| 0-30 | 680 | GSOCmap v1.5 | (FAO and ITPS, 2018) |
| | 1267 | SoilGrids250m | (FAO and ITPS, 2018) |
| | 755±119 | WISE30sec | (Batjes, 2015) |
| | 699 | HWSD V1.1 | (Hiederer and Köchy, 2011) |
| | 504 | WISE5by5min | (Hiederer and Köchy, 2011) |
| | 574 | DSMW | (Hiederer and Köchy, 2011) |
| | 684-724 | WISE | (Batjes, 1996) |

To further improve the overall accuracy of soil data, future work is expected to narrow the circles in Figure 8 and bring them closer to the upper right corner. First of all, it is essential to establish a unified and standardized system for soil sampling, measurement, recording and calculation methods in the global SOC simulation, which requires global soil science communities to collaborate and work collectively (Onerhime, 2021). Secondly, data sources need to be populated with all possible efforts to appropriately cover different regions and biomes (Fig. 8 and S13). The potential solutions can be region-specific and biome-

specific. For example, in the Northern circumpolar region (Fig. 8), or for forest and wetlands (Fig. S13), the sampling density of soil profiles could be increased, and soil profile data sharing among the various soil databases should be promoted given the inherent difficulties of sampling and measuring in this region (Mishra et al., 2013). In the regions (e.g., China, and North America) (Fig. 8) or biomes (e.g., croplands) (Fig. S13) with higher density of observations, soil datasets can be summarized and integrated to form a more accurate database. In the regions with large differences among field observations, using more

reliable data sources and enhancing soil sampling campaigns are necessary to reduce the uncertainty in SOC estimation. Last

but not the least, data sharing should be further encouraged along the line of dataset development, e.g., during planning, sampling, measuring, cross-validation, mapping at either individual, regional or national level (Lobry de Bruyn and Ingram, 2019). Citizen science, and cross-program (programs with needs for soil sampling) collaborations should also be pursued as options for soil data collection (Rossiter et al., 2015; Robinson et al., 2019).

**5 Conclusions**

Global SOC density and stocks vary greatly among different datasets. The overall stocks are at a range of 577-1171 Pg and 1086-2678 Pg with averages of 828 and 1873 Pg, for 0-30 cm 0-100 cm soil depths, respectively. In terms of spatial distribution, the SOC density is higher in the high latitudes and equatorial regions, and the SOC density gradually increases from low latitude to high latitude. Globally, GSOCmap provides relatively accurate estimate of SOC stocks at 0-30 cm depth, and SoilGrids250m

and GSDE perform better for simulations at multiple depths. At the regional scale, these five SOC estimates have various accuracies in different areas, and the largest differences are in the northern circumpolar and Tibetan Plateau permafrost regions. The UM2021 and WISE30sec perform better in the northern circumpolar permafrost regions, and GSDE performs better in China. For different biomes, the five soil datasets show better performance in simulating SOC density of grasslands, croplands and shrublands/savannas, and perform poorly over permanent wetlands. The uncertainty in SOC estimates mainly comes from

soil sampling density, diverse sources and mapping methods, as well as the interpolation methods used by different authors. In the future, to reduce the uncertainty in SOC estimates, the sample density of soil profiles should be increased in areas and biomes with large SOC uncertainties, and global cooperation and data sharing should be further encouraged.

**Data availability**

We made all the data used in this study publicly accessible to assist reduce the uncertainty in global and regional SOC estimates.

This dataset includes 7 TIFF files for 0-30 cm SOC density from the five global soil datasets and the regional permafrost dataset (spatial resolution: 30″, units: t ha$^{-1}$) and 6 TIFF files for 0-100 cm SOC density from the four global soil datasets and the regional permafrost dataset (spatial resolution: 30″, units: t ha$^{-1}$). The free availability of this dataset does not imply free publication. Any use of this dataset should include appropriate acknowledgement to the original data sources. This dataset is available at https://doi.org/10.6084/m9.figshare.20220234 (Lin et al., 2022). The information provided in Table S2 lists links

for downloading global and regional datasets used in this study.

**Supplement.**

The supplement related to this article is available online at: …

**Author contributions.**

Z. Q. conceived the idea and designed the study. Z. L. and Z. Q. collected and analyzed the datasets, and Y. D., U. M, G. W., W.
Z., and W. S. helped with interpretation of the results. Y. D. and W. S. provided help with GSDE dataset, U. M. helped with
UM2021 data access. Z. L. and Z. Q. wrote the main manuscript with contributions from all authors. All authors reviewed and
edited the manuscript.

**Competing interests.**

The authors declare that they have no conflict of interests.

**Acknowledgements.**

 We are very grateful to Dr. Giulio Genova for communicating on soil product use of SoilGrid250m, and other authors and
developers of the global and regional soil datasets for sharing their knowledge and products.

**Financial support.**

This work has been partially supported by the National Natural Science Foundation of China (U21A6001, 41975113), and the
Guangdong Provincial Department of Science and Technology (2019ZT08G090).

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
