# Peer review of "On the magnitude and uncertainties of global and regional soil organic carbon: A comparative analysis using multiple estimates"

_Earth System Science Data, 2022_

## Referee Comment (RC1)

Review ESSD-2022-232 SOC

Does ESSD publish reviews of global data sets? Evidently: yes? I find global population estimates (https://doi.org/10.5194/essd-11-1385-2019) and global CO2 emissions (https://doi.org/10.5194/essd-12-1437-2020). A competent review of global SOC could certainly qualify.

Authors here face problem composed of individual problems. Hard enough to quantify SOC across terrain and biome in local region or national region. Then to combine those estimates into systematic harmonized global estimates? Double difficult! Then to review such products? Much needed, certainly, but - unfortunately - not well done here. Disappointment results from dismal outcomes (SOC across products varies by at least a factor of two or three) or from dismal approach? These authors conclude (line 267) "no consensus on global SOC estimates in these observation-based estimates". This review accepts that outcome but evaluates how well authors have produced and backed it up.

Products show greatest variation (discrepancies) in northern and high-altitude permafrost. No surprise. Permafrost community presents those discrepancies persistently. These authors set up their permafrost product, named UM2021, for "further comparison" with regional "most up-to-date SOC" as 'reference' but, in text, basically added UM2021 as only an additional data source. They also set up WoSIS for comparison/reference but, as most users will know, WoSIS profiles, covering very different depths, present soil texture, density and water-holding physical parameters inconsistently and - also inconsistently - soil bioochemical pararmeters such as SOC (e.g https://doi.org/10.5194/essd-12-299-2020, Batjes cited here). All efforts within WoSIS to ensure quality and document uncertainty (e.g. in WoSIS 2019 update) seem lost here? WoSIS not listed in e.g. their Table 1, but shows up as reference point in every Taylor plot? What appropriate subset of WoSIS did these authors use? Authors have not assured readers on what basis they applied WoSIS? Finally, they describe satellite remote sensing vegetation product (MODIS, so-called MCD12C1) but that gets early mention then disappears; no mention (other than reference) after line 137 (did authors mean MCD12C1 or MCD21C1?). Very disappointing and, unfortunately, very limiting in how they estimate (or not) any uncertainties nor attempt any validation.

Three of their five products represent basically FAO products, e.g. assembled and harmonized by FAO from national reports to FAO. But, no FAO co-author nor citation of FAO literature?

All analysis occurs basically on spatial basis, complicated by (too-simple) depth info (30 cm vs 100 cm). Spatially, SOC concentrations vary. So? Didn't we all know that beforehand. One product better in one region, another better in another region; no surprise in any of this. Spatial patterns vary for 0-100 cm vs 0-30 cm. Again, no surprise. As authors conclude (line 280): "diversity of data sources and different mapping methods are important reasons for the differences". But, if at least three products derive from same underlying data, how does 'diversity' emerge? What uncertainties arise from original materials, from assembly and harmonization process, during interpolation (to gridded fields)? Not one hint here. How do spatial patterns in any product fit remote sensing vegetation patterns. Again, no hint. Figure 9 attempts to document workflow, but: a) many acronyms cited in Fig 9 appear nowhere else in manuscript; b) reader gets no hint about apparent schemes (e.g. panel a, b, etc.); c) co-variates listed in Fig 9 appear receive only brief mention and gain minimal assessment elsewhere in manuscript; and d) no uncertainties estimated or validations approximated at any step identified in Fig. 9.

At lines 140,141 authors tell readers that water bodies "not discussed in this study". But, authors use and cite "permanent wetlands" as one of their representative biomes. From SOC

viewpoint, how do they use one but not the other? One might extract a soil profile from a wetland but not from a lake?

Biggest concern has to do with complete thorough absence of any discussion of temporal variation. Authors cite heterotrophic respiration (line 34) as important consumption term for SOC, but neglect that entirely in subsequent analysis. Readers and authors will know that heterotrophic respiration (e.g. SOC conversion to $CH_4$ or $CO_2$) varies with temperature, substrate, soil moisture, etc. In other words, seasonally as well as at longer (decadal) time scales. Not one hint from these authors about which, if any, of these global SOC products support temporal analysis. If, as this reviewer contends, one can not, and must not, attempt to interpret SOC spatial patterns in absence (or, in ignorance) of temporal patterns, how do any of these products withstand serious temporal scrutiny? If no global SOC product supports high fidelity temporal analysis, tell us so? Climate use requires temporal variability terms!

Overall, this reviewer finds spatial conclusions unsurprising, absence of uncertainties limiting, lack of any attempt at validation disappointing, and failure to at least mention temporal variation unacceptable.

Could most readers have completed this analysis and summary? No. From point of view of carbon cycles, do we need accurate SOC numbers starting from assessment of currently available global products? Yes. Do readers encounter an inclusive and rigorous approach? Unfortunately, no.